# "Energetics of the outer retina I: Estimates of nutrient exchange and ATP generation"

Stella Prins[1,2], Christina Kiel[3], Alexander J. E. Foss[4], Moussa A. Zouache[5], Philip J. Luthert[1,6]*

1 UCL Institute of Ophthalmology, London, United Kingdom, 2 Advanced Research Computing Centre, University College London, London, United Kingdom, 3 Department of Molecular Medicine, University of Pavia, Pavia, Italy, 4 Department of Ophthalmology, Nottingham University Hospitals NHS Trust, Nottingham, United Kingdom, 5 John A. Moran Eye Center, University of Utah, Salt Lake City, Utah, United States of America, 6 NIHR Moorfields Biomedical Research Centre, UCL Institute of Ophthalmology, London, United Kingdom

* p.luthert@ucl.ac.uk

## Abstract

Photoreceptors (PRs) are metabolically demanding and packed at high density, which presents a challenge for nutrient exchange between the associated vascular beds and the tissue. Motivated by the ambition to understand the constraints under which PRs function, in this study we have drawn together diverse physiological and anatomical data in order to generate estimates of the rates of ATP production per mm$^2$ of retinal surface area. With the predictions of metabolic demand in the companion paper, we seek to develop an integrated energy budget for the outer retina. It is known that rod PR number and the extent of the choriocapillaris (CC) vascular network that supports PRs both decline with age. To set the outer retina energy budget in the context of aging we demonstrate how, at different eccentricities, decline CC density is more than matched by rod loss in a way that tends to preserve nutrient exchange per rod. Together these finds provide an integrated framework for the study of outer retinal metabolism and how it might change with age.

**Data Availability Statement:** There are no primary data in this study and all relevant data sources are cited. All the code used in the metabolic simulation can be found at https://figshare.com/articles/

## Introduction

Photoreceptors (PRs), like most excitable cells, require substantial energy flows to function normally and in addition are packed together at a remarkably high spatial density as part of the biological adaptations that optimise visual acuity. This presents a major challenge in terms of nutrient supply and exchange and, in most vertebrates, this is addressed through a planar vascular bed that lies parallel and posterior to the light-sensing outer segments (OSs) of rods and cones. Between this vascular bed, which is known as the choriocapillaris (CC), and the PRs lies a connective tissue layer (Bruch's membrane) and a monolayer of cells, the retinal pigment epithelium (RPE) (S1 Fig in S1 File). Average CC blood flow is remarkably high, but the functional topography of this vascular bed means that there is significant heterogeneity of flow, which in some areas may be limiting [1–3].

software/Human1_RPE-PR/26764219?file=48623152.

**Funding:** This work was supported by a grant to PJL and CK from Moorfields Eye Charity (https://moorfieldseyecharity.org.uk - Grant GR001345) and a BBSRC (Biotechnology and Biological Science Research Council - https://www.ukri.org/councils/bbsrc/) award (BB/N003616/1) to PL. The funders played no role in the study.

**Competing interests:** The authors have declared that no competing interests exist.

In this study and in the companion paper (Kiel C, Prins S, Foss AJE, et al. [submitted]) we seek to integrate knowledge from multiple sources to develop a comprehensive, integrated model of outer retinal bioenergetics. In this context, the outer retina complex includes the CC, RPE and PRs external to the outer limiting membrane. Bioenergetics embraces the understanding of how RPE and PR cells convert nutrients into usable energy (mainly in the form of ATP) and how, in turn, ATP is used to drive cellular processes. Here we focus on nutrient exchange and the capacity of the RPE and PRs to generate ATP; that is the supply side of the energy budget. In the companion study we provide estimates of energy demand, in terms of ATP hydrolysis, drawing together physiological data relating to the dynamics of processes carried out by the RPE and PRs and estimates of cellular capacity to carry out specific processes informed by enzyme or transporter abundance and turnover rates (Kiel C, Prins S, Foss AJE, et al. [submitted]).

The formulation of an integrated energy budget, where supply and demand are fully specified and quantified, provides the foundational basis for a systems understanding of how energy flows through the tissue and back into the circulation as well as a framework for defining fundamental questions about the health and disease of the outer retina. Such a framework would help address understanding of issues such as how demands of multiple processes are balanced against one another; whether or not there is competition between different cell types for nutrients and how heterogeneities of supply are dealt with. Another issue in disease states is the energetic impact of the homeostatic processes evoked to deal with a pathological perturbation or 'insult' to the system. Finally, vascular supply and tissue demand alter as part of the aging process [4]. The interplay of supply and demand may underlie how in some individuals, aging is associated with preservation of retinal function but in others, profound visual impairment can develop. In this study we explore how in healthy aging there are concurrent shifts in both supply and demand, which may be critical in maintenance of homeostasis of the aging eye.

To inform studies of human disease, we have built our model around the human eye, although this has required extrapolation from various sources of animal data including those from physiological studies and investigations of single-cell gene expression.

The inaccessibility of the back of the eye makes direct, *in vivo*, measurement of nutrient fluxes extremely challenging if not impossible, but they are central to the understanding of the energy flows that maintain tissue integrity and visual function. The organisation of the vascular supply also makes it difficult to measure arteriovenous (AV) differences in metabolite concentrations in many mammalian species of interest. It is, however, possible to make such measurements in the cat and in this study we take advantage of seminal experiments exploring glucose uptake and lactate efflux in the cat [5]. Linsenmeier and colleagues have successfully measured oxygen tension profiles across the retina and used these to derive estimates of PR oxygen uptake, [6, 7] which we have also utilised. The relative density of rods and cones varies markedly with increasing distance from the central retina, known as the fovea. To explore how supply per PR varies with retinal eccentricity and with age we have drawn on landmark anatomical studies [8–10] and more recent imaging studies using optical coherence tomography angiography (OCTA) [11].

To create a model of outer retinal metabolism where nutrient exchange is linked to ATP generation (as the main currency of energy flow), we adopted cell specific genome-scale metabolic models with metabolite exchange between choroidal and retinal circulations and the tissue and with rod and cone models coupled to a cell-specific model of the retinal pigment epithelium.

## Materials and methods

### Cell-specific metabolic models

Genome-scale metabolic models (GEMs) represent metabolic reaction networks and enable simulation and analysis of metabolic processes [12]. They contain information about gene-protein-reaction relationships (GPRs), which map gene products to specific reactions and include rules that, for instance, manage multiple enzyme isoforms and complexes. This enables integration of gene expression data (as a proxy for protein expression) to create customised models that reflect distinct cellular contexts and physiological states. In this study, we utilized the latest and most comprehensive human GEM to date, Human1 [13]. Cell-specific models were created using CORDA2, an implementation of Cost Optimization Reaction Dependency Assessment [14–16] (CORDA).

Before the CORDA2 algorithm was applied, gene expression data were mapped onto the reactions. First, single-cell expression data from two datasets—one containing RPE data [17, 18] and one containing rod and cone PR data [19], were discretized into the following five categories: not present (values $< .00001$), low confidence (values between 0.0001 and the first quartile), medium confidence (values between the first and third quartiles), high confidence (values above the third quartile), and unknown confidence (missing values). Gene symbols used in the datasets were matched to the Ensembl Gene IDs used in Human1. Employing functions from the CORDA package, expression confidence levels were linked to reactions through the GPRs. After this, the CORDA2 algorithm was used to reconstruct cell-specific GEMs. The biomass equation (a single expression that captures the net metabolic requirements of reproducing the macromolecular components of a cell) was set as objective function to eliminate reactions within pathways that do not contribute to biomass production. The code for coupling the RPE and PR models together with the cell-specific models is available at https://figshare.com/articles/software/Human1_RPE-PR/26764219?file=48623152.

### Coupling of RPE and PR metabolic models

To capture the complexity of retinal metabolism, we coupled the cell-specific RPE and PR metabolic models (S1 Fig in S1 File). Before coupling the models, the ATP hydrolysis reaction (Human1 reaction ID MAR03964) and any exchange reactions that were lost during the creation of the cell-specific models were added back. Metabolite and reaction IDs were altered by adding suffixes '_RPE' or '_PR'. An RPE-PR ([e_RPE_PR]) interface was created by duplicating reactions involving the extracellular space compartment and giving them unique metabolite and reaction ID suffixes; 'e_RPE_PR' for the extracellular metabolites, '_PR_RPE' for the reactions in the PR model, and '_RPE-PR' for the reactions in the RPE model. After this, models were fused, duplicated reactions deleted, and reaction ID suffixes of reactions only involving the e_RPE_PR compartment changed to '_eRPE_PR'. The models were saved in SBML format.

### Blood exchange constraints

A key metabolic constraint of any tissue is the rate of exchange of key metabolites between the circulation and the extracellular space (S1 Fig in S1 File). In the retina, a unique vascular arrangement involves a dual circulation system: the inner retina is supplied by branches from the central retinal artery, while the outer retina relies primarily on the CC for blood supply. To account for this, our coupled RPE-PR model has two interfaces between with the external environment—one with the extracellular space of the PRs ([e_PR]) and the other on with the extracellular space of the RPE cells ([e_RPE])—allowing the exchange of metabolites.

## Oxygen

Oxygen delivery is a critical determinant of a tissue's capacity to generate ATP because oxidative phosphorylation depends on it. The most reliable data regarding oxygen uptake in the outer retina come from measurements of oxygen partial pressure profiles. These measurements have been performed in various species, with the macaque being the species most similar to humans, for which such data are available [6, 7]. For the oxygen influx estimates in this study we used published macaque data for the perifoveal and foveal regions in light and dark [6]; 2.46 and 3.74 pmol·s$^{-1}$·mm$^{-2}$ respectively for the perifoveal region, and 2.34 and 2.72 pmol·s$^{-1}$·mm$^{-2}$ respectively for the foveal region. As previously mentioned, the retina receives its blood supply from two circulatory systems: the CC and the retinal circulation. While there is no retinal circulation in the foveal pit, in the perifoveal region the relative contribution of the inner retina supply system to PR oxygen uptake is 15% in the dark and 11% in the light [7]. We represented this in our model by splitting the perifoveal oxygen uptake in two components: oxygen uptake by the [e_RPE] and oxygen uptake by the [e_PR]; 2.19 and 0.27 pmol·s$^{-1}$·mm$^{-2}$ respectively (light), and 3.18 and 0.56 pmol·s$^{-1}$·mm$^{-2}$ respectively (dark). Transfer of oxygen from the extracellular RPE-PR interface compartment, [e_RPE_PR], to the PR cytosol compartment, [c_PR], was fixed (see S2 Table in S1 File).

The mathematical approach taken in the oxygen profile measurements estimates PR oxygen uptake and does not provide oxygen consumption rates for the RPE. To account for this, we added 0.30 pmol·s$^{-1}$·mm$^{-2}$ to the oxygen uptake rate by the RPE. This value was informed by two studies. In the first by Calton and colleagues [20], RPE oxygen consumption rates were ~ 100 pmol·min$^{-1}$ in triangular patches of RPE cells (see their Figure 2C [20]). These patches had sides with a length (*a*) of 3.15 mm. To calculate the flux per unit area we treated the RPE patches as equilateral triangles with a surface area of $A_{triangle} = \sqrt{3}/4a^2$. This allowed us to estimate that the oxygen flux per unit area of these cells was about 0.31 pmol·s$^{-1}$·mm$^{-2}$. In the second study by Kurihara and colleagues [21], the RPE oxygen consumption rate was around 200 pmol·min$^{-1}$ (see their Figure 7D). Using the well surface area of the Agilent 96-well XF plate (11 mm$^2$; https://www.agilent.com/cs/library/images1/5991-8747EN_seahorse_microplates_schematic.pdf), we calculated a flux per unit area of about 0.30 pmol·s$^{-1}$·mm$^{-2}$. While there are variations in the literature regarding RPE oxygen uptake estimates—differences possibly influenced by degree of RPE differentiation or culture conditions—our estimates are in line with the upper range of values found in other studies [21].

It is curious that the estimated foveal oxygen flux is lower than that in the perifoveal region even though it might be predicted that the high density of cones in the foveal region would require a higher flux to meet energy demands [22]. To explore the theoretical maximum oxygen flux, we used Fick's first law of diffusion. Estimates of the distance from the CC to the inner aspect of the IS / ellipsoid zone (IS/EZ) line, were extracted from high resolution OCT images [23]. The perifoveal distance is approximately 58 μm. Taking the diffusion coefficient of oxygen to be 1.97 x 10$^{-5}$ cm$^2$·s$^{-1}$, the solubility of oxygen in blood to be around 138 pmol·mmHg$^{-1}$, an oxygen tension of 100 mmHg in the CC, a CC area fraction for exchange of 0.7, and an assumption of zero oxygen tension at the inner aspect of the IS/EZ line, we get a theoretical maximum oxygen flux of 3.29 pmol·s$^{-1}$·mm$^{-2}$, which is close to our estimate above of 3.74. Interestingly, at the centre of the fovea the IS/EZ line is further from the CC at 77 μm with an associated flux of 2.47 pmol·s$^{-1}$·mm$^{-2}$ which again is close to the above estimate of 2.72. The Fick's estimates don't include the component of IS oxygen supply that comes from the inner retina. A further consideration is that the lipid-rich environment of the RPE and PRs elevates the diffusion constant for oxygen [24]. For further analysis we have elected to use the values derived from the oxygen profile measurements.

## Glucose and lactate

To enhance the physiological fidelity of our model, we also set boundaries for the exchange of glucose and lactate, two key metabolites in tissue metabolism. The bounds for glucose and lactate were informed by published AV difference measurements for the cat choroid in the light and dark [5]. The cat retina has a very high peak spatial density of rods [25] and in this regard is a reasonable model of the human retina, which also has a high peak spatial density of rods [8]. To convert total retinal AV differences to flux per $mm^2$ we took the average cat eye radius ($r$) to be approximately 11 mm [26] and we estimated the proportion of eye surface occupied by the retina to be 65%. Treating the cat eye as a perfect sphere gives an estimate of the surface area of the cat retina of $0.65 \cdot 4\pi r^2$. As an aside, the cat choroid oxygen uptake rates per unit area that we could now estimate from the AV differences in oxygen concentration in the same paper (S1 Table in S1 File), are strikingly similar to the macaque estimates (S2 Table in S1 File). Because the cat oxygen AV differences will include a contribution to RPE metabolism, the RPE does not have to be considered separately in this case. We made the assumption that the ratio of oxygen consumption to glucose uptake and lactate efflux in the cat in the light and the dark can be extrapolated to the macaque and hence human so taking the macaque oxygen flux data as the reference we kept the ratios of oxygen flux to those of the other metabolites, in light and dark, constant (S2 Table in S1 File).

## Amino acids

It is known that amino acid uptake by the RPE is important [27, 28] and we sought to incorporate amino acid fluxes into the model. As AV difference data for amino acids were not available, we set the bounds of uptake relative to that for glucose in the ratio of glucose to plasma concentration for each amino acid (S3 Table in S1 File). Plasma concentrations of glucose and the amino acids were taken from the Human Metabolome Database (https://hmdb.ca/) [29]. Specific kinetics of amino acid uptake and competition at transporters were not modelled.

## Other metabolites

All other metabolites are free to leave the system through two interfaces with the circulation. Additionally, water is allowed to enter the system freely.

## ATP yield

To assess maximal ATP yield in different conditions, classical flux balance analysis (FBA) was performed using COBRApy (https://opencobra.github.io/cobrapy/) with the objective function set to maximise ATP hydrolysis in the PR (MAR03964_PR: ATP[c_PR] + $H_2O$[c_PR] $\Leftrightarrow$ ADP[c_PR] + $H^+$[c_PR] + $P_i$[c_PR]). This is equivalent to the rate of ATP production or ATP yield. Where flux estimates for intermediate reactions were studied flux variability analysis, loopless and parsimonious FBA from COBRApy were used.

## Age-related structural changes in CC and rod densities

Rod spatial density measurements were taken from Figures 7A and 11A of Curcio et al 1993 [9] and values corrected for the authors' estimates of specimen expansion. The four age groups provide values of rod density from the centre of the retina to an eccentricity of 5 mm for individuals in approximately 3rd, 4th, 7th and 9th decades. To facilitate arithmetic operations between densities at different ages, polynomial fits to the data were established using R (version 4.2.3), which was also used for other analyses.

Regression line coefficients of variation of CC density with age [10] were used and related to age-related changes in flow void measurements from an OCTA study [11]. CC density in this context is the proportion of a line passing along the CC that contains blood vessels in a histological section. This length per unit length measure provides an estimate of the area fraction of the CC where the line can be considered randomly oriented in the plane of the CC. This seems to be a reasonable assumption close to the macula.

## Results

### Estimates of ATP yield

Cell-specific models of the metabolic pathways of RPE, rod PR, and cone PR were created using the Human1 database [13] as the basis and removing reactions according to gene expression data using the CORDA2 algorithm [15, 16]. RPE-cone PR and RPE-rod PR coupled models were then constructed, to capture the metabolic environment in the outer retina. As a control, instead of cell-specific models, two template Human1 models were linked. Flux balance analysis (FBA) was used to determine the maximal rate of ATP yield in the PR under oxygen, glucose and lactate blood-tissue exchange constraints and in light and dark conditions (S3 Table in S1 File) that was estimated from published experimental data (see Methods).

### ATP yield in light and dark

Under dark conditions, with glucose as the only available fuel (left panels, Fig 1), the maximal rate of perifoveal PR ATP synthesis is 28.5 pmol·s$^{-1}$·mm$^{-2}$ for the control model and 27.9 pmol·s$^{-1}$·mm$^{-2}$ for retina-specific models. Under the same conditions, foveal PR ATP yield was 17.54 pmol·s$^{-1}$·mm$^{-2}$ for the control model and 15.5 pmol·s$^{-1}$·mm$^{-2}$ for retina-specific

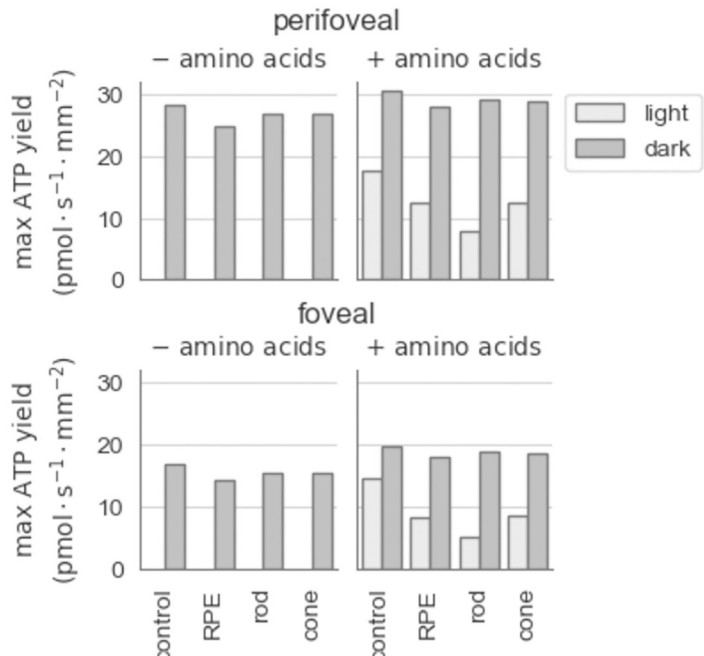

**Fig 1. Maximal ATP yield during light and dark.** The maximal ATP yield in pmol·s$^{-1}$·mm$^{-2}$ during light (light grey) and dark (dark grey), in absence (left panels) and presence (right panels) of amino acids for the control model and for the retina-specific models. The control model contains two coupled Human1 models, while the retina-specific models consist of an RPE model coupled to a PR rod ('rod') or cone ('cone') model.

models. There were no feasible solutions for any of the models when the light conditions were imposed with glucose as the only source of carbons so we explored the possible impact of allowing amino acids to enter from the circulatory compartment. In the light, with amino acids included, the maximal rate of perifoveal ATP yield was 17.8 pmol·s$^{-1}$·mm$^{-2}$ for the control model and 17.59 pmol·s$^{-1}$·mm$^{-2}$ for the retina-specific models. Foveal ATP yield in these conditions was 16.34 pmol·s$^{-1}$·mm$^{-2}$ and 16.14 pmol·s$^{-1}$·mm$^{-2}$ for the control and retina-specific models respectively (right panels, Fig 1). The inclusion of amino acid uptake on the model in the dark led to a small increase in the maximum ATP yield. Specifically, perifoveal ATP yield increased to 30.7 pmol·s$^{-1}$·mm$^{-2}$ for the control model and 30.05 pmol·s$^{-1}$·mm$^{-2}$ for the retina-specific models and foveal ATP yield increased to 21.61 pmol·s$^{-1}$·mm$^{-2}$ and 21.22 pmol·s$^{-1}$·mm$^{-2}$ for the control and retina-specific models respectively.

## The effect of varying magnitude of maximal amino acid uptake

We tested the effect of variations in the maximal allowed rate of amino acid influx by running the model amino acids at double, half, one fourth of the originally estimated rate (S2 Fig; S4 Table in S1 File). Doubling the upper limit of amino acid influx results in only a marginal increase in ATP production. For the control model, the increase in ATP yield ranged from 0.51 to 1.19 pmol·s$^{-1}$·mm$^{-2}$, while for retina-specific models, it was between 0.32 and 0.57 pmol·s$^{-1}$·mm$^{-2}$. Under light conditions, reducing amino acid influx has a more pronounced impact on retina-specific models compared to the control model. More specifically, in retina-specific models, both perifoveal and foveal ATP production decrease to less than half of their initial values when amino acid availability is halved. Notably, in retina-specific models under light conditions, reducing amino acid influx to a quarter of the original values failed to yield any feasible solution for ATP production as the extra carbons are needed to meet the magnitude of lactate efflux determined from experimental studies [5].

## Varying glucose influx

To determine the minimal glucose influx required to still allow a feasible solution, glucose availability was reduced in decrements of 0.10 pmol·s$^{-1}$·mm$^{-2}$ (S3 and S4 Figs in S1 File). As can be seen in S5 Table in S1 File, for the control model, the minimum foveal glucose requirements under light conditions were 1.6 pmol·s$^{-1}$·mm$^{-2}$ without amino acids and 1.0 pmol·s$^{-1}$·mm$^{-2}$ with amino acids. Under light conditions, the retina-specific models had the same foveal glucose requirements when amino acids were introduced, however without amino acids the glucose requirements were slightly higher; 1.2 pmol·s$^{-1}$·mm$^{-2}$. In dark conditions, the foveal glucose requirements of the retina-specific models further differentiate from control. Here, the minimum foveal glucose requirements were 2.5 pmol·s$^{-1}$·mm$^{-2}$ without and 1.4 pmol·s$^{-1}$·mm$^{-2}$ with amino acids for the control model, and 2.6 pmol·s$^{-1}$·mm$^{-2}$ without and 1.9 pmol·s$^{-1}$·mm$^{-2}$ with amino acids for the retina-specific models. The perifoveal region (S4 Fig in S1 File) has higher glucose requirements; 2.1 (light) and 4.2 (dark) pmol·s$^{-1}$·mm$^{-2}$ without amino acids, 1.3 (light) and 2.3 (dark) pmol·s$^{-1}$·mm$^{-2}$ with amino acids for the control condition, 2.4 (light) and 4.6 (dark) pmol·s$^{-1}$·mm$^{-2}$ without amino acids, 1.7 (light) and 3.1 (dark) pmol·s$^{-1}$·mm$^{-2}$ with amino acids for the retina-specific models.

## Varying lactate efflux

As shown in S5 Table in S1 File, in the control model the threshold for foveal lactate efflux rates (S5 Fig in S1 File) under light conditions is 3.0 pmol·s$^{-1}$·mm$^{-2}$ when amino acids were absent, and a higher rate of 4.2 pmol·s$^{-1}$·mm$^{-2}$ in their presence. During light exposure, the retina-specific models, had a slightly lower lactate efflux threshold; 2.9 pmol·s$^{-1}$·mm$^{-2}$ without amino

acids and 3.7 pmol·s$^{-1}$·mm$^{-2}$ when supplemented with amino acids. In dark conditions the foveal lactate efflux thresholds were 5.7 pmol·s$^{-1}$·mm$^{-2}$ without amino acids in all models, with amino acids it increased to 8.1 pmol·s$^{-1}$·mm$^{-2}$ for the control model, and to 7.1 pmol·s$^{-1}$·mm$^{-2}$ for the retina-specific models. The perifoveal region has higher lactate efflux thresholds (S6 Fig in S1 File). For the control model under light conditions, lactate efflux was 4.0 pmol·s$^{-1}$·mm$^{-2}$ without amino acids and 5.7 pmol·s$^{-1}$·mm$^{-2}$ with amino acids. For retina-specific models, the rates were 3.6 pmol·s$^{-1}$·mm$^{-2}$ without amino acids and 4.9 pmol·s$^{-1}$·mm$^{-2}$ with them. In dark conditions the perifoveal lactate efflux thresholds in the control model were 9.6 pmol·s$^{-1}$·mm$^{-2}$ without amino acids and 13.4 pmol·s$^{-1}$·mm$^{-2}$ with amino acids. Again, retina-specific models had slightly lower efflux rates of 8.9 pmol·s$^{-1}$·mm$^{-2}$ without amino acids and 11.9 pmol·s$^{-1}$·mm$^{-2}$ when they were present.

## Varying oxygen influx

The effect of varying oxygen influx rates is shown in S7 and S8 Figs in S1 File. At both foveal and perifoveal sites, with the exception of simulations with amino acids and in the dark, the CORDA2-constrained models show a reduction in ATP yield at higher oxygen fluxes. Analysis of the flux vectors from simulations at different oxygen fluxes show that at high oxygen fluxes there is export of hydrogen peroxide back into the circulation.

## Inclusion of lipids

It is well-recognised that PRs metabolise lipids and it seems highly likely that this is at least in part lipid arising from recycling of PR OSs. To explore how the metabolic network can utilize lipids to generate ATP, simulations were conducted using the control network (Human1). We assessed the impact of four lipid pools: extracellular phospholipids, VLDL-associated fatty acids, fatty acid uptake, and the general fatty acid pool, corresponding to Human1 reaction IDs MAR09089, MAR13037, MAR13039, and MAR09209, respectively. Under conditions simulating the dark perifoveal environment, we set lactate, glucose, and oxygen exchange to their experimental values. Only the fatty acid uptake pool (MAR13039, S6 Table in S1 File) affected ATP production. Setting the influx bounds to 1 pmol·s$^{-1}$·mm$^{-2}$, increased the maximal ATP yield from 28.3 to 29.4 pmol·s$^{-1}$·mm$^{-2}$. No further increase in ATP yield was observed when lipid availability was further increased. To further explore the capacity of lipid availability to affect ATP yield, the lipid availability in the fatty acid-uptake pool was varied against different lactate efflux rates, and glucose and oxygen availabilities, as shown in S9 Fig in S1 File. Given the minimal impact of lipids, further analysis of cell-specific models was not carried out.

## Variation of nutrient exchange per PR with increasing eccentricity and age

The flux estimates considered above are averages for what we assume can be considered young adult experimental animals, but in the human, there are marked differences in rod and cone density at increasing distances from the foveal centre of the retina and there are also changes in CC and PR density with age. If we assume that nutrient exchange between the CC and PRs is one-dimensional, that is only operates perpendicular to the plane of the CC, then the area fraction of the CC and the cross-sectional area of the PR inner segment (IS), will both be key determinants of exchange per PR. The fractional area of the CCs available for exchange has not been examined systematically with increasing eccentricity although for the elderly (mean age 82.4 y), data are available for central retina (within an approximately 1 mm diameter circle) and outside of this area (i.e. extramacular) where CC area fractions or densities have been reported as 0.51 ± 0.08 and 0.55 ± 0.6 (mean ± SD) respectively [10]. There are, however, data

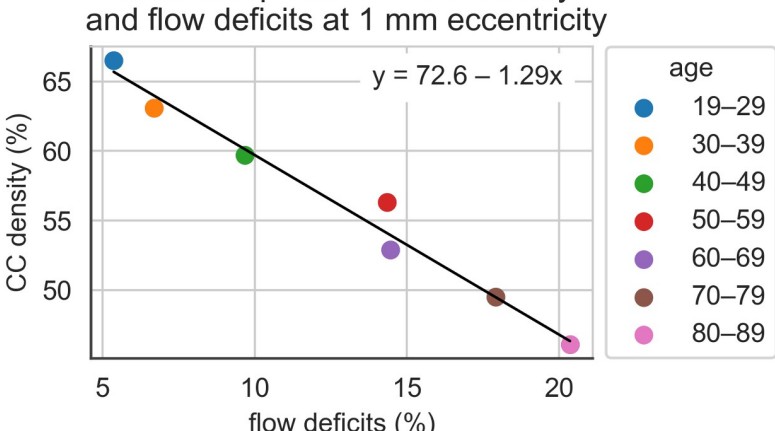

**Fig 2. Scatterplot depicting the inverse correlation between the percentage of flow deficits from OCTA data (Zheng et al., 2019) and CC density across different age ranges at 1 mm eccentricity from the fovea.** The fitted regression line (y = 72.6 – 1.29x) illustrates the amount at which CC density decreases as flow deficits increase. CC density was estimated using regression line coefficients of variation of CC density with age, published by Ramrattan and colleagues [10].

available from OCTA studies that cover the central 5 mm of the retina and also show changes with age [11].

To establish the relationship between OCTA flow voids and CC area fraction (or density), data derived from the Ramrattan histology study [10] for each decade were plotted against those from the OCTA investigation for the same decade. There was an inverse linear relationship with an adjusted $r^2$ of 0.972 (Fig 2). We used this relationship to translate the OCTA flow void data to more physiologically relevant anatomical measures of the CC. Using the calibration in Fig 2 we have converted OCTA flow deficit data (Fig 3A) to CC densities (Fig 3B).

Taking the mean CC density in the youngest age group as a baseline, if we assume that nutrient exchange is proportional to the area of CC in a plane parallel with Bruch's membrane, the fractional change in CC density with varying age and eccentricity (Fig 3B) can be used to estimate shifts in nutrient exchange. Note that the OCTA data informing Fig 3B are only

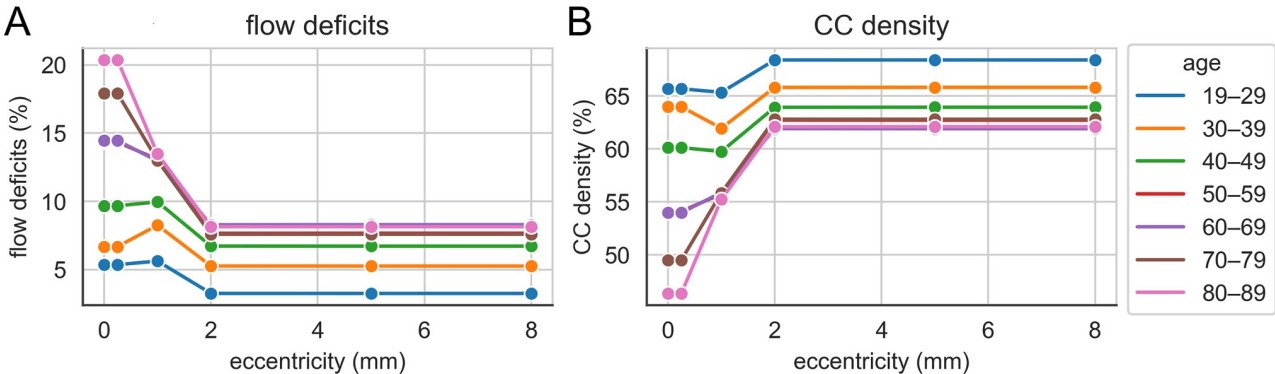

**Fig 3. Retinal vascular parameters by age and eccentricity.** (**A**) Percentage of flow deficits in retinal vessels as detected by OCTA scans from Zheng et al., 2019, categorized by age and measured at various distances from the foveal centre. (**B**) Predicted CC density at different eccentricities estimated using the regression model from Fig 1. Age groups are indicated by color-coded lines and markers.

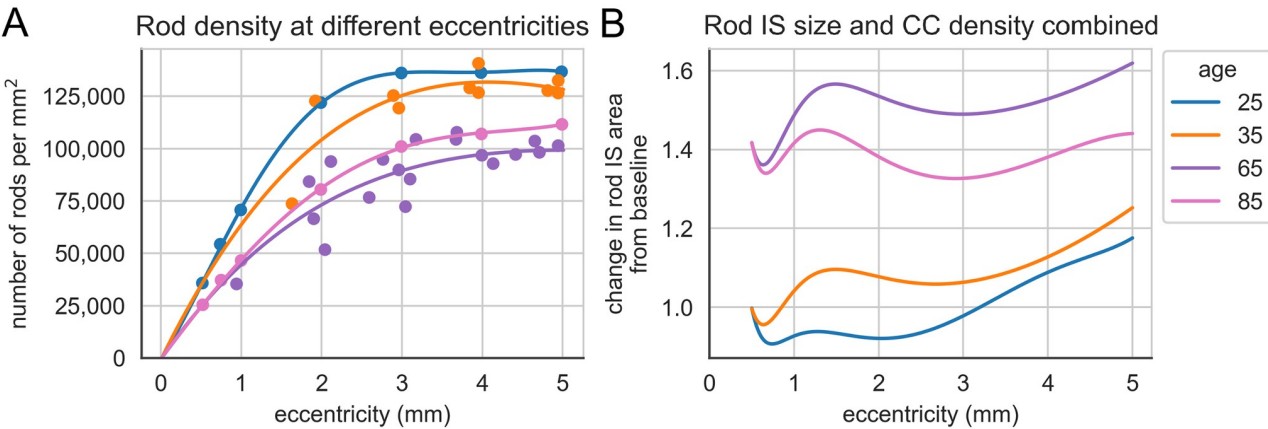

**Fig 4. Retinal changes with age and eccentricity.** (**A**) Distribution of rod PR density, comparing data for four distinct age groups based on findings from Curcio et al., 1993 (specifically Figure 7A and Figure 11A). (**B**) Fractional changes in rod IS cross-sectional area in 4 different age groups and at varying eccentricity. Data at small eccentricities are unreliable as cones predominate and rounding errors and variability between studies become problematic.

available at a limited number of eccentricities and that simple linear interpolation has been used and the most central value is assumed to hold to the fovea and similarly, the most peripheral value is used further to the periphery.

Rod density also changes with eccentricity and with age and plots derived from [9] are shown in Fig 4A. Note that although in general there is progressive loss of rods with age the oldest group has higher density than the third oldest group, possibly reflecting the degree of individual variation in PR number. By subtracting the area occupied by cones and dividing by number of rods per unit area, the IS cross-sectional area of the rods can be estimated given that it appears as though as rods are lost with age the cross sectional area of the IS of the remaining rods enlarge [9]. They are therefore able to 'capture' more nutrient exchange. Taking the '25' year group as baseline, ignoring the central 0.5 mm segment from the foveal centre (as the cones are dominant and rounding errors between different datasets become problematic) it is possible to look at the fractional change in rod area with again, varying eccentricity and with age (Fig 4B).

It can be seen that the fractional changes in CC and rod area work in opposite directions during aging with enlarging rods capturing more nutrients and loss of CC reducing nutrient availability. This is shown diagrammatically in S10 Fig in S1 File. When the combined effects are plotted (taking the product of the two fractional changes at each age and location, see Fig 5) it is apparent that the increased size of rods more than compensates for the degenerative changes in the CC in the eldest two groups. There does, however, appear to be a zone of relative paucity of supply to rods in the youngest group from about 0.6 to 3 mm eccentricity.

## Discussion

There is increasing availability of biomedical data [30], yet it remains a challenge to generate insights into mechanism [31]. Here we have sought to address this challenge by integrating diverse published data to develop a model of human outer retinal nutrient exchange and capacity to generate ATP as well as how supply may vary with increasing age and at different distances from the fovea. This analysis has been carried out both in the foveal region and more peripherally in the perifovea. At the fovea, the oxygen flux is lower with a resulting lower ATP yield.

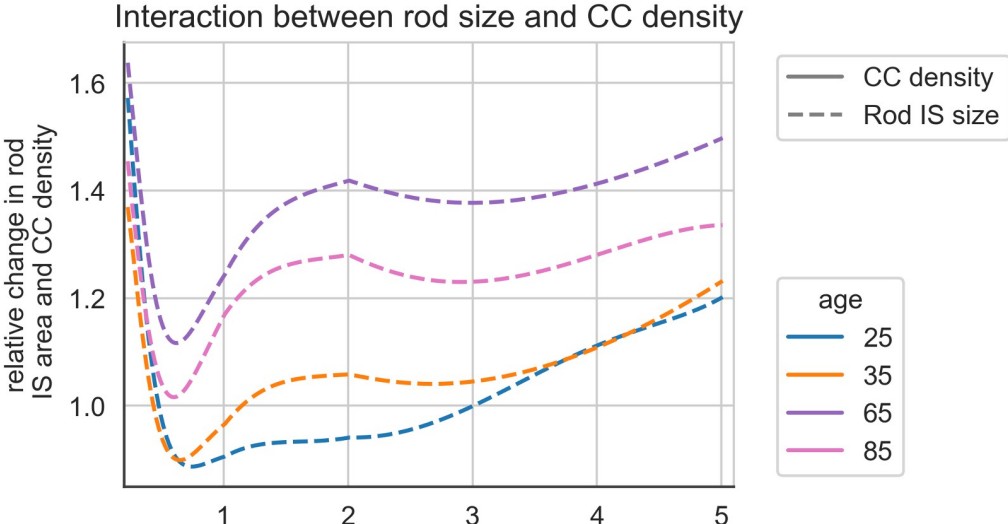

**Fig 5. Graph showing how fractional change in flux varies with age and eccentricity.** The solid lines give the changes predicted from altered CC density. When these changes are combined with those in rod cross sectional area (Fig 4B) the resulting shifts are shown with dotted lines.

One novel aspect of this work is the use of OCTA imaging to derive parameters to inform modelling of physiology. It appears possible to translate OCTA CC flow void data into an anatomical measure of vessel density (Fig 1) and also, to take the distance from the CC to inner aspect of the IS/EZ line as an estimate of maximum rate of oxygen flux to ISs from the CC (see Methods). This approach may find utility in personalised approaches to retinal conditions.

The correlation between mRNA and protein abundance is a contentious issue, with studies showing varying degrees of correlation [32]. This challenge is exacerbated by the fact that most RNA-seq and proteomics datasets are unpaired and conducted under different conditions [33]. In this work, the CORDA method was employed to create RPE, rod- and cone-specific models, mitigating some of these uncertainties. Rather than using precise RNA levels, RNA sequencing data was categorised into distinct confidence levels to determine whether or not specific reactions should be included in the network. When it becomes available, integration of proteomic data to refine network reaction boundaries has the potential to improve the biological fidelity of the models.

There seems to be minimal loss of ATP generating capacity in these cell-specific models compared to the control 'unspecified' model. Further work incorporating additional constraints imposed by relative level of expression of different metabolic enzymes and metabolite transporters will likely expose greater differences between cell types.

We used experimental findings on retinal oxygen exchange from macaques, along with data on oxygen, glucose, and lactate exchange in cats, to constrain our models. While macaque eyes closely resemble human eyes and provide the most relevant data available, it's important to acknowledge the differences in cat eyes. A key difference is that cats lack a fovea. While the human fovea is exclusively composed of cone photoreceptors, cats possess an area centralis characterized by a peak cone density that is still about ten times lower than the density of rods in this area [34]. Consequently, applying our foveal findings to humans requires careful consideration.

The model with the imposed exchange constraints and with glucose as the only carbon source, did not provide a feasible solution with data obtained in the light. As can be seen in the

parameter sweeps, relaxing the constraints around either glucose uptake or lactate efflux led to a feasible solution. One possible explanation is that the system is finely balanced and measurement error is responsible. The small AV differences in metabolite concentrations present a technical challenge [5]. It was, however, of interest that when amino acid uptake was allowed to take place, a solution was possible. Whether or not amino acid uptake is essential in vivo warrants further investigation but it is noteworthy that there are experimental data showing significant uptake of amino acids [21, 27, 35]. A further possible source of carbons for oxidative phosphorylation is lipid. It is certainly known that PRs can metabolise lipid but it is less clear how much of this is derived from the circulation and how much from turnover of OS-associated lipid. Simply opening up the model to circulating lipid only increases ATP yield slightly as might be predicted from the relatively low oxygen flux.

It is interesting that there is marked variability of CC $pO_2$ in the profile measurements (see [7], Figure 4). Although the spatial resolution of the microelectrodes is considered to be of the order of 1 micron we speculate that the CC and RPE signals may be hard to distinguish and that there may be high RPE oxygen uptake that can't be discriminated. Hurley and colleagues have recently shown that intermediaries from the tricarboxylic acid (TCA) cycle may be exchanged between PR and ISs enabling IS ATP production but with the final oxygen-consuming step in the RPE [36]. If this was the case, supply could be greater than we predict as we would have under-estimated RPE $O_2$ consumption.

A challenge in understanding how biological systems age comes from the potential complexity of interactions between different tissue elements. It seems logical that with age-related cell death, in this instance of rod PRs, there is reduction in metabolite demand, but age-related degradation of the microvasculature might drive PR cell loss. Here we have shown that in the absence of disease it appears as though the increased ability of aged rods to capture metabolites exceeds the degree to which the CC fails. There is, however, in the youngest group, a suggestion of relatively reduced nutrient flux from about 0.6 to 3 mm eccentricity. It is possibly not a coincidence that this is the region of maximum rate of rod loss (see [9], Figure 7B). So there remains the possibility that the pattern of rod loss arises as a function of impaired nutrient exchange and bioenergetics in the young and that there is an overshoot in loss which, in the elderly, more than compensates for the impaired supply. This formulation might reconcile the apparent paradox that maximum rod loss is central to the region of maximum rod density where the lower density of cones and the greater integrity of the CC means the nutrient exchange per rod is higher. More centrally, rods seem better served despite the degradation of the CC under the very centre of the retina. Possibly this offers an explanation for foveal sparing seen in some degenerative disorders of the retina including age-related macular degeneration [37], mitochondrial retinopathies [38] and other retinal degenerations [39], although this may simply be a function of high cone and low rod density in the fovea.

The companion paper (Kiel C, Prins S, Foss AJE, et al. [submitted]) provides estimates of ATP demand and, together with the supply estimates here, makes it possible to consider an RPE and PR IS energy budgets for the cone-rich fovea and rod-rich perifoveal retina (Table 1).

**Table 1. ATP supply and demand in pmol·s⁻¹·mm⁻².**

|  |  | Supply | Demand |
|---|---|---|---|
| Foveal | Light | 16.14 | 11.21 |
|  | Dark | 21.22 | 10.64 |
| Perifoveal | Light | 17.59 | 10.84 |
|  | Dark | 30.05 | 28.04 |

On average, our estimates of supply are 1.4 times higher than estimates of demand with a particularly close match in the rod-rich perifoveal tissue in the dark. There are many potential sources of error that may contribute to this discrepancy. For instance, there will be measurement errors in the data we have based the modelling on; FBA optimises ATP yield and assumes the system is 100% efficient, which is probably not the case. Indeed, there are data suggesting that photoreceptor mitochondria are uncoupled to an unusual degree, which would reduce ATP production whilst oxygen fluxes were maintained [40, 41]. There may also be demands on ATP that we are not aware of or have under-estimated. The synthesis of data from different species is not ideal and the metabolic model used lacks the fidelity of a fully-specified, 3D, kinetic model. It would also be preferable to have longitudinal datasets of rod and cone densities at different eccentricities with functional assessment of CC blood flow and more detailed, real-time, local measures of metabolite exchange. We have assumed an average capacity for nutrient exchange at different eccentricities, which is likely an over-simplification. Nevertheless, the establishment of an initial energy budget for the outer retina provides a framework for further studies. The current pace of ocular imaging advances will likely assist in taking some of the concepts outlined here forward. Given the importance of understanding the drivers of degeneration in the macula, further studies exploring the hypotheses that emerge from this study seem warranted.

There are many additional limitations to this study and indeed, we consider it to be a first step towards a high fidelity, integrated description of the bioenergetics of the human retina and associated tissues. Flux balance analysis, as employed here, only captures steady-state phenomena, whereas in reality, the retina is a dynamic system responding swiftly to, for instance, changes in illumination, which will bring specific metabolic challenges [42], that are not addressed by the current approach. Model validation is an important challenge but, as here, the direct measurements that would be required for validation are often not possible with current methodologies and the modelling can be seen as hypothesis-generating or as offering feasible solutions but not direct demonstration of what is taking place. The ultimate aim of this work is to be able to predict evolution of changes with age and disease, but this is well beyond the scope of this current study. Validation of such a model would itself require clinical determination of its predictive capability.

## Supporting information

**S1 File.**
(DOCX)

## Acknowledgments

We are most grateful to Jamie Quinn (UCL Advanced Research Computing) for assistance with programming.

## Author Contributions

**Conceptualization:** Christina Kiel, Alexander J. E. Foss, Moussa A. Zouache, Philip J. Luthert.

**Data curation:** Stella Prins.

**Funding acquisition:** Christina Kiel, Philip J. Luthert.

**Software:** Stella Prins, Philip J. Luthert.

**Writing – original draft:** Stella Prins, Philip J. Luthert.

**Writing – review & editing:** Stella Prins, Christina Kiel, Alexander J. E. Foss, Moussa A. Zouache, Philip J. Luthert.

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
