## [Decision Letter · Decision Letter 0]

25 Jul 2024

PONE-D-24-23589“Energetics of the outer retina I: Estimates of nutrient exchange and ATP generation.”PLOS ONE

Dear Dr. Luthert,

Thank you for submitting your manuscript to PLOS ONE. After careful consideration, we feel that it has merit but does not fully meet PLOS ONE’s publication criteria as it currently stands. Therefore, we invite you to submit a revised version of the manuscript that addresses the points raised during the review process. **Specifically, it would be useful to compare the actual levels of expression of PI kinases and phosphatases in rods, cones and RPE cells reported in previous studies with the levels of RNA transcripts reported from the Iowa RNA seq studies in GEM analyses, as suggested by the reviewer 2. **

We look forward to receiving your revised manuscript.

Kind regards,

Alexandre Hiroaki Kihara, Ph.D.

Academic Editor

PLOS ONE

Journal Requirements:

This work was supported by a grant to PJL and CK from Moorfields Eye Charity (https://moorfieldseyecharity.org.uk - Grant GR001345) and a BBSRC (Biotechnology and Biological Science Research Council - https://www.ukri.org/councils/bbsrc/) award (BB/N003616/1) to PL. The funders played no role in the study.

The authors gratefully acknowledge the support of Moorfields Eye Charity (Grant GR001345), the BBSRC (BB/N003616/1) and NIHR Moorfields Biomedical Research Centre. We are also most grateful to Jamie Quinn (UCL Advanced Research Computing) for assistance with programming

This work was supported by a grant to PJL and CK from Moorfields Eye Charity (https://moorfieldseyecharity.org.uk - Grant GR001345) and a BBSRC (Biotechnology and Biological Science Research Council - https://www.ukri.org/councils/bbsrc/) award (BB/N003616/1) to PL. The funders played no role in the study.

5. We noted in your submission details that a portion of your manuscript may have been presented or published elsewhere. A pre-print of this paper was submitted to BioRxiv as part of the submission to PLOS Computational Biology. Please clarify whether this [conference proceeding or publication] was peer-reviewed and formally published. If this work was previously peer-reviewed and published, in the cover letter please provide the reason that this work does not constitute dual publication and should be included in the current manuscript.

6. Thank you for uploading your study's underlying data set. Unfortunately, the repository you have noted in your Data Availability statement does not qualify as an acceptable data repository according to PLOS's standards.

7. PLOS requires an ORCID iD for the corresponding author in Editorial Manager on papers submitted after December 6th, 2016. Please ensure that you have an ORCID iD and that it is validated in Editorial Manager. To do this, go to ‘Update my Information’ (in the upper left-hand corner of the main menu), and click on the Fetch/Validate link next to the ORCID field. This will take you to the ORCID site and allow you to create a new iD or authenticate a pre-existing iD in Editorial Manager. Please see the following video for instructions on linking an ORCID iD to your Editorial Manager account: https://www.youtube.com/watch?v=_xcclfuvtxQ

8. We note that you have referenced "Kiel C, Prins S, Foss AJE, et al." which has currently not yet been accepted for publication. Please remove this from your References and amend this to state in the body of your manuscript: (Kiel C, Prins S, Foss AJE, et al. [submitted]”) as detailed online in our guide for authors

Reviewers' comments:

Reviewer's Responses to Questions

**Comments to the Author**

1. Is the manuscript technically sound, and do the data support the conclusions?

Reviewer #1: Yes

Reviewer #2: Yes

2. Has the statistical analysis been performed appropriately and rigorously? 

Reviewer #1: N/A

Reviewer #2: N/A

3. Have the authors made all data underlying the findings in their manuscript fully available?

Reviewer #1: Yes

Reviewer #2: Yes

4. Is the manuscript presented in an intelligible fashion and written in standard English?

Reviewer #1: Yes

Reviewer #2: Yes

5. Review Comments to the Author

**Reviewer #1:** In this study the authors aimed to explore how in healthy aging there are concurrent shifts in both supply and demand, which may be critical in maintenance of homeostasis of the aging eye.

The study heavily relies on computational simulations and extrapolations from animal data due to the challenges of directly measuring nutrient fluxes and metabolic activities in the human retina. While efforts were made to validate through available experimental data (such as oxygen tension profiles), direct validation of the entire metabolic model in vivo remains challenging.

The study provides insights into static metabolic states (light vs. dark conditions), but retinal metabolism is dynamic and can vary under different physiological and pathological states (e.g., aging, disease). The model's ability to capture these dynamic changes is limited by current data and assumptions.

While the study lays a foundation for understanding normal retinal bioenergetics, translating these findings to clinical applications (e.g., understanding retinal diseases or therapeutic strategies) requires further validation and adaptation to human clinical context

The study's exploration of age-related changes in retinal bioenergetics highlights complexities in metabolic interactions between different retinal layers and their vascular supply. However, understanding how these interactions evolve over time, especially in the context of disease-related metabolic shifts, warrants further investigation beyond the study's current scope.

**Reviewer #2:** Strengths:

The authors used a genome-scale metabolic computational program to mathematically model the production of metabolic energy in rod and cone photoreceptors and in the retinal pigment epithelium. The GEM modeling program (Human1) and constraint-based models are established and widely used so I think this now is considered by many to be an important way to initiate a comprehensive overview of metabolism in many types of cells. The authors also use data from oxygen consumption measurements to constrain their analyses and measurements of glucose and lactate in the arteries and veins leading to and from a mammalian retina. The major strength of this report is that it is a pioneering study of this type of computational analysis of the retina and RPE. It is important because it will serve as a starting point for future analyses that steadily improve on the modeling as more and more data become available. Analyses of what happens when expression of specific enzymes are blocked and comparison with data from actual metabolic analyses of animals in which those genes are inactivated will be very useful in future studies. Overall, the authors describe their strategies and findings clearly in this manuscript. The analyses used single-cell RNA sequencing data that I think most investigators consider to be reliable.

Weaknesses:

I think it would be surprising that a model based on transcript abundances could accurately predict the actual abundances of the relevant proteins and their activities. It may be possible for the authors to address this. There is published data that might provide a way to test the validity of the assumption that transcript abundance is proportional to protein abundance and activity. There is a published analysis of expression of phosphoinositide kinase and phosphatase proteins in RPE and photoreceptor cells that was reported by Rajala et al. (PMID 37007713 and 35427794). Those papers quantified the expression of PI kinases and phosphatases in specific cell types by isolating ribosomes specifically either from RPE cells, rods, cones or other cells. It would be useful for the authors (Prins et al.) to compare the actual levels of expression of those proteins (PI kinases and phosphatases) in rods, cones and RPE cells reported in those studies with the levels of RNA transcripts reported from the Iowa RNA seq studies that they used in their GEM analyses. If transcript and protein abundances are in good agreement with each other then the authors could use that to strengthen the interpretation of their analyses. If they do not agree then the authors should still report their findings but they should include for their readers a discussion of the extent to which quantifying transcripts rather than the actual amounts of enzymes and their activities adds uncertainty to the interpretation.

Reviewer: James B. Hurley

6. PLOS authors have the option to publish the peer review history of their article (what does this mean?). If published, this will include your full peer review and any attached files.

Reviewer #1: **Yes: **Hamid Riazi-Esfahani

Reviewer #2: **Yes: **James B. Hurley

---

## [Author Response · Author response to Decision Letter 0]

27 Aug 2024

Response to Reviews Comments

Reviewer #1: In this study the authors aimed to explore how in healthy aging there are concurrent shifts in both supply and demand, which may be critical in maintenance of homeostasis of the aging eye.

Comment 1: “The study heavily relies on computational simulations and extrapolations from animal data due to the challenges of directly measuring nutrient fluxes and metabolic activities in the human retina. While efforts were made to validate through available experimental data (such as oxygen tension profiles), direct validation of the entire metabolic model in vivo remains challenging.”

Reply 1: This is an important point. Part of the classical cycle of computational modelling involves validation of model predictions that were not ‘baked in’ to the original model (see Figure 1 in 1). Unfortunately, as we mention (lines 88–95), many of the key parameters of interest in relation to human outer retinal metabolism in vivo are simply inaccessible to current experimental methods. So, in this instance we are offering plausible hypotheses as to what might be going on rather than providing definitive proof. The hope is that this work will at least in part help provoke development of technologies that can validate our findings.

We have strengthened the description of study limitations in the discussion (lines 592-594).

Comment 2: “The study provides insights into static metabolic states (light vs. dark conditions), but retinal metabolism is dynamic and can vary under different physiological and pathological states (e.g., aging, disease). The model's ability to capture these dynamic changes is limited by current data and assumptions.”

Reply 2: We have indeed only modelled static steady states. Dynamic changes, which are of great interest, require a fully parametrised kinetic model. This is a major undertaking and we are working on this but realistically we are 1 – 2 years away. Other groups may be closer. One of the benefits of the current study is that it provides steady-state points against which to compare a dynamic model.

We have articulated this thinking in the revised discussion and provided a reference to a publication that discusses metabolic dynamics of the retina (lines 594-7).

Comment 3: “While the study lays a foundation for understanding normal retinal bioenergetics, translating these findings to clinical applications (e.g., understanding retinal diseases or therapeutic strategies) requires further validation and adaptation to human clinical context”

Reply 3: We completely agree. The current work is the first step towards gaining deeper understanding of diseases and therapeutics. Many further steps will be required to achieve the clinical impact we ultimately seek to achieve.

We have included this towards the end of the revised discussion (lines 601-4).

Comment 4: “The study's exploration of age-related changes in retinal bioenergetics highlights complexities in metabolic interactions between different retinal layers and their vascular supply. However, understanding how these interactions evolve over time, especially in the context of disease-related metabolic shifts, warrants further investigation beyond the study's current scope.”

Reply 4: We agree.

A statement to this effect has been incorporated into the revised discussion (lines 601-3).

Reviewer #2: Strengths:

Comment 5: “The authors used a genome-scale metabolic computational program to mathematically model the production of metabolic energy in rod and cone photoreceptors and in the retinal pigment epithelium. The GEM modeling program (Human1) and constraint-based models are established and widely used so I think this now is considered by many to be an important way to initiate a comprehensive overview of metabolism in many types of cells. The authors also use data from oxygen consumption measurements to constrain their analyses and measurements of glucose and lactate in the arteries and veins leading to and from a mammalian retina. The major strength of this report is that it is a pioneering study of this type of computational analysis of the retina and RPE. It is important because it will serve as a starting point for future analyses that steadily improve on the modeling as more and more data become available. Analyses of what happens when expression of specific enzymes are blocked and comparison with data from actual metabolic analyses of animals in which those genes are inactivated will be very useful in future studies. Overall, the authors describe their strategies and findings clearly in this manuscript. The analyses used single-cell RNA sequencing data that I think most investigators consider to be reliable.”

Reply 5: We are grateful for the encouragement! We also see this work as a starting point from which we and other groups can build models of increasing biological fidelity.

We have emphasised this in the revised discussion (lines 592-4).

Weaknesses:

Comment 6: “I think it would be surprising that a model based on transcript abundances could accurately predict the actual abundances of the relevant proteins and their activities. It may be possible for the authors to address this. There is published data that might provide a way to test the validity of the assumption that transcript abundance is proportional to protein abundance and activity. There is a published analysis of expression of phosphoinositide kinase and phosphatase proteins in RPE and photoreceptor cells that was reported by Rajala et al. (PMID 37007713 and 35427794). Those papers quantified the expression of PI kinases and phosphatases in specific cell types by isolating ribosomes specifically either from RPE cells, rods, cones or other cells. It would be useful for the authors (Prins et al.) to compare the actual levels of expression of those proteins (PI kinases and phosphatases) in rods, cones and RPE cells reported in those studies with the levels of RNA transcripts reported from the Iowa RNA seq studies that they used in their GEM analyses. If transcript and protein abundances are in good agreement with each other then the authors could use that to strengthen the interpretation of their analyses. If they do not agree then the authors should still report their findings but they should include for their readers a discussion of the extent to which quantifying transcripts rather than the actual amounts of enzymes and their activities adds uncertainty to the interpretation.”

Reply 6: We are grateful for being directed to the interesting PI kinase and phosphatase papers, which will doubtless inform our future studies. From our reading, the methodology used involves assaying actively transcribed mRNA and therefore the data do not represent actual protein abundances). 

There is certainly an important issue when seeking to extrapolate from mRNA to protein abundancies. The correlation is often poor. As others have pointed out2 , while there is an abundance of RNA-seq and proteomics data available, the majority of these datasets are unpaired with RNA-seq and proteomics conducted under different conditions. Given the varying dynamic ranges and technical biases across different mRNA and protein assays, merging of various datasets might potentially magnify the divergence between mRNA and protein levels.

There have been several approaches to address this problem in GEM studies. The CORDA method used in this work, doesn’t hinge on precise expression levels. Rather, it categorizes expression data into five distinct levels: unknown, not detected, low confidence, medium confidence, and high confidence. The expression confidence levels are mapped on the reactions after which the algorithm includes all high confidence reactions and maximizes the inclusion of medium confidence reactions, while minimizing the incorporation of non-present reactions. This approach determines whether or not reactions can occur, rather than setting the precise magnitude boundaries of the reactions. This strategy mitigates the impact of the uncertainty between mRNA and protein levels. While the correlation between mRNA and protein levels may be weak, the CORDA method allows for a reasonable representation of the system’s state.

We completely agree that the use of protein abundances would be optimal and hope that in future studies it will be possible to integrate proteomic and immunohistochemical data into network definition and flux constraints or rates.

We have amplified the discussion relating to the challenges of using mRNA as opposed to protein abundances and given additional references (lines 490-499).

Comment from companion paper Kiel et al

Comment 32: “3. lines 347-353. It would be helpful to discuss the importance of mitochondrial coupling. Photoreceptor mitochondria may be more uncoupled than in most other types of cells. The authors cite reference 46 (in paper 2), which is based on measurements of mitochondria from cultured cells and from mitoplast preparation. The properties of mitochondria can vary in the degree to which they are uncoupled and photoreceptor mitochondria may be particularly unique in this way. This should be discussed.” 

Response 32: The issue of mitochondrial uncoupling is important and we completely agree that photoreceptor mitochondria may be unique! Oxygen consumption rate doesn’t increase greatly following uncoupling 3, which could reflect basal activity being very high, as in a cardiomyocyte 4. But we agree that the limited drop in OCR with oligomycin may well reflect a significant degree of basal uncoupling. Nevertheless, curiously, and perhaps reflecting that the effect isn’t massive, inhibiting oxidative phosphorylation results in a significant drop in ATP production for the retina (see Fig 1 in the Cepko group’s paper 5). It’s not clear how much of this drop relates to photoreceptors and how much to inner retina.

We have added a comment about this (lines 576-8), emphasising that uncoupling may explain why our estimates of ATP production are higher than our estimates of demand. We have also added references to Dr Hurley’s work on this topic6, 7, to illustrate the point.

1. Roberts PA, Gaffney EA, Luthert PJ, et al. Mathematical and computational models of the retina in health, development and disease. Prog Retin Eye Res 2016; 53: 48-69. 20160407. DOI: 10.1016/j.preteyeres.2016.04.001.

2. Prabahar A, Zamora R, Barclay D, et al. Unraveling the complex relationship between mRNA and protein abundances: a machine learning-based approach for imputing protein levels from RNA-seq data. NAR Genom Bioinform 2024; 6: lqae019. 20240210. DOI: 10.1093/nargab/lqae019.

3. Kooragayala K, Gotoh N, Cogliati T, et al. Quantification of Oxygen Consumption in Retina Ex Vivo Demonstrates Limited Reserve Capacity of Photoreceptor Mitochondria. Invest Ophthalmol Vis Sci 2015; 56: 8428-8436. 2016/01/10. DOI: 10.1167/iovs.15-17901.

4. Mootha VK, Arai AE and Balaban RS. Maximum oxidative phosphorylation capacity of the mammalian heart. Am J Physiol 1997; 272: H769-775. DOI: 10.1152/ajpheart.1997.272.2.H769.

5. Chinchore Y, Begaj T, Wu D, et al. Glycolytic reliance promotes anabolism in photoreceptors. Elife 2017; 6 20170609. DOI: 10.7554/eLife.25946.

6. Hass DT, Bisbach CM, Sadilek M, et al. Aerobic Glycolysis in Photoreceptors Supports Energy Demand in the Absence of Mitochondrial Coupling. Adv Exp Med Biol 2023; 1415: 435-441. DOI: 10.1007/978-3-031-27681-1_64.

7. Du J, Rountree A, Cleghorn WM, et al. Phototransduction Influences Metabolic Flux and Nucleotide Metabolism in Mouse Retina. J Biol Chem 2016; 291: 4698-4710. 20151216. DOI: 10.1074/jbc.M115.698985.

---

## [Decision Letter · Decision Letter 1]

2 Oct 2024

“Energetics of the outer retina I: Estimates of nutrient exchange and ATP generation.”

PONE-D-24-23589R1

Dear Dr. Luthert,

We’re pleased to inform you that your manuscript has been judged scientifically suitable for publication and will be formally accepted for publication once it meets all outstanding technical requirements.

Kind regards,

Alexandre Hiroaki Kihara, Ph.D.

Academic Editor

PLOS ONE

Additional Editor Comments (optional):

Reviewers' comments:

Reviewer's Responses to Questions

**Comments to the Author**

1. If the authors have adequately addressed your comments raised in a previous round of review and you feel that this manuscript is now acceptable for publication, you may indicate that here to bypass the “Comments to the Author” section, enter your conflict of interest statement in the “Confidential to Editor” section, and submit your "Accept" recommendation.

Reviewer #2: All comments have been addressed

2. Is the manuscript technically sound, and do the data support the conclusions?

Reviewer #2: (No Response)

3. Has the statistical analysis been performed appropriately and rigorously? 

Reviewer #2: Yes

4. Have the authors made all data underlying the findings in their manuscript fully available?

Reviewer #2: Yes

5. Is the manuscript presented in an intelligible fashion and written in standard English?

Reviewer #2: Yes

6. Review Comments to the Author

Reviewer #2: (No Response)

7. PLOS authors have the option to publish the peer review history of their article (what does this mean?). If published, this will include your full peer review and any attached files.

Reviewer #2: **Yes: **James Bryant Hurley

---

## [Editor Report · Acceptance letter]

15 Oct 2024

PONE-D-24-23589R1 

PLOS ONE

Dear Dr. Luthert, 

I'm pleased to inform you that your manuscript has been deemed suitable for publication in PLOS ONE. Congratulations! Your manuscript is now being handed over to our production team.

Kind regards, 

on behalf of

Dr. Alexandre Hiroaki Kihara 

Academic Editor

PLOS ONE
